# Bayesian U-Net: Estimating Uncertainty in Semantic Segmentation of Earth Observation Images

**Clément Dechesne [1,*], Pierre Lassalle [1] and Sébastien Lefèvre [2]**

[1]  CNES, 18 Avenue Edouard Belin, 31401 Toulouse, France; pierre.lassalle@cnes.fr
[2]  UMR CNRS 6074, Campus de Tohannic, University Bretagne Sud–IRISA, 56000 Vannes, France; sebastien.lefevre@irisa.fr
[*]  Correspondence: clement.dechesne@cnes.fr

**Abstract:** In recent years, numerous deep learning techniques have been proposed to tackle the semantic segmentation of aerial and satellite images, increase trust in the leaderboards of main scientific contests and represent the current state-of-the-art. Nevertheless, despite their promising results, these state-of-the-art techniques are still unable to provide results with the level of accuracy sought in real applications, i.e., in operational settings. Thus, it is mandatory to qualify these segmentation results and estimate the uncertainty brought about by a deep network. In this work, we address uncertainty estimations in semantic segmentation. To do this, we relied on a Bayesian deep learning method, based on Monte Carlo Dropout, which allows us to derive uncertainty metrics along with the semantic segmentation. Built on the most widespread U-Net architecture, our model achieves semantic segmentation with high accuracy on several state-of-the-art datasets. More importantly, uncertainty maps are also derived from our model. While they allow for the performance of a sounder qualitative evaluation of the segmentation results, they also include valuable information to improve the reference databases.

**Keywords:** deep neural network; semantic segmentation; bayesian network; optical imagery; uncertainty estimation

## 1. Introduction

Deep learning is considered one of the major breakthroughs related to big data and computer vision [1]. It has been extensively applied to semantic segmentation using fully convolutional networks for pixelwise prediction [2] or combined with deconvolution (encoder–decoder architecture) to segment objects of different sizes within the same image [3]. The encoder–decoder architecture has been widely used for semantic segmentation [4]. Indeed, the most popular frameworks for semantic segmentation rely on such an encoder–decoder architecture, e.g., U-Net [5] or Segnet [6]. Such frameworks have been widely used for the semantic segmentation of optical images [7–11] with very high accuracies. The methods proposed by [7,9] used several Fully Convolutional Neural Networks (FCN) or Convolutional Neural Networks (CNN) for semantic segmentation on aerial orthophotos with three spectral bands (red, green, near-infrared), plus a digital surface model (DSM) of the same resolution. They both reported excellent results for a five-class classification task (*roads, buildings, low vegetation, trees, cars*) with an overall accuracy greater than 88%, and efficient detection of small objects (such as individual cars). In [8], in addition to the FCNN, a boundary detection CNN module is added, increasing the accuracy of the model. The framework proposed by [11] used a refinement module in their FCNN, trained on six-band airborne images for an 18-class classification task, achieving good results with an overall accuracy greater than 93%. They also showed that data augmentation is meaningful for semantic segmentation. In [12], a Dense U-Net is used to perform semantic segmentation on the Vaihingen dataset proposed by the International Society for Photogrammetry and Remote Sensing (ISPRS) and also tackle the problem

of class-unbalance. They obtain good results, showing that the U-Net architecture with focal loss is relevant to semantic segmentation and the detection of small objects. The method proposed by [13] uses a Residual U-Net for the semantic segmentation of remotely sensed data. It uses a U-Net encoder/decoder backbone, in combination with residual connections. The performance of the Generalized Dice loss for semantic segmentation is analyzed, and a novel variant loss function is introduced for the semantic segmentation of objects. The performance of this framework is evaluated on the ISPRS 2D Potsdam dataset, showing a state-of-the-art performance with an average F1 score of 92.9% over all classes.

Despite their successes, traditional deep neural networks still have several limitations. They most often perform tasks by learning through examples without prior knowledge of the tasks. Millions of parameters have to be learned thanks to the optimization process, usually with stochastic gradient descent. Convolutional neural networks have already surpassed human accuracy in many vision tasks. Due to the convolutional neural networks' ability to fit a wide diversity of non-linear datapoints, they require a large amount of training data. Furthermore, neural networks are generally prone to overfitting on small datasets. The model tends to fit the training data well, but is not accurate for new data. This often makes the neural networks incapable of correctly assessing the uncertainty in the training data, and hence leads to overly confident decisions. In order to avoid over-fitting, several regularization techniques have been proposed, such as *early stopping*, *weight decay* or *L1 and L2 regularizations*. Currently, the most popular and empirically effective technique to reduce over-fitting is *dropout* [14,15].

Although Neural Networks architectures have achieved state-of-the-art results in almost all classification tasks, they still make over-confident decisions. No measure of uncertainty is provided from the current Neural Network architectures. A few works have been proposed to generate relevant probability estimates from a deep neural network [16,17] as a measure of model confidence. There are also popular metrics, such as the expected calibration error and the maximum calibration error, which can be used to quantitatively measure model calibration. However, these metrics are based on softmax probabilities, which cannot capture epistemic or model uncertainty [18]. Furthermore, there are very simple post-processing techniques, such as temperature scaling [16], which can make a deterministic and probabilistic model equally relevant. An approach has been proposed to add weight uncertainty [19], providing an uncertainty measure for deep learning models. Combined with variational inference [20], Bayesian learning can be applied to Convolutional Neural Networks by adding both a measure of uncertainty and weight regularization to their predictions [21].

Bayesian Deep learning has been proposed for semantic segmentation, to provide uncertainty in the prediction. It can be seen as a forest of deep neural networks, with each providing a single prediction. It has been shown [22] that dropout (initially designed to avoid overfitting [14,15]) can be used as a Bayesian approximation. This method, called Monte Carlo (MC) Dropout, has been applied [23] for the semantic segmentation of the Cityscape dataset [24]. They designed a DeepLab model [25] with MC Dropout and achieved great results, with an overall accuracy of 95.3% and Intersection over Union (IoU) of 78%. They also provide, along with the semantic segmentation output, several uncertainty maps (namely *predictive entropy* and *mutual information*), showing that the model uncertain of its prediction for pixels where the prediction is erroneous. MC Dropout has also been applied [26] to a SegNet [6] architecture. The SegNet was trained on CamVid Road Scenes [27] and SUN RGB-D Indoor Scene Understanding [28] datasets. It achieved better results than the state-of-the-art methods and also provided uncertainty maps (for all classes and per class, based on output variability). Reference [29] improved traditional MC dropout, slightly increasing the accuracy of the model. More recently, Reference [30] compared MC Dropout to another Bayesian Deep model, where weights are no longer single values but are sampled from a distribution. In this case, the model learns the parameters of the distribution instead of the weights. They showed that such models produce better results and more interpretable uncertainty maps. However, these models

require significant training times. Reference [10] proposes CNN models that showed success in the semantic segmentation of high-resolution optical images. They showed that uncertainty maps, derived from Monte Carlo dropout [22], allow for the classification accuracy to be improved from 90% to 97.5% by removing uncertain pixels from prediction. The framework proposed by [31] also applied Monte Carlo dropout to RADARSAT SAR images for the segmentation of roads. They showed that such a strategy would be helpful in the semi-automated road segmentation frameworks using Deep Learning on SAR images.

In this paper, we tackle the problem of uncertainty estimation in Earth Observation (EO) semantic segmentation. Earth observation (EO) is the gathering of information about the physical, chemical, and biological systems of the planet Earth. Earth observation is used to monitor and assess the status of changes in the natural and built environments. The uncertainty estimation is obtained by coupling the most widespread architecture in the field (namely U-Net) with the Bayesian learning framework. This strategy allows us to obtain not only the fully convolutional network trained on EO data, but also maps of uncertainty, which indicate places where the network was not able to make predictions with significant confidence. In this work, we do not aim to produce a new semantic segmentation method. We prefer to rely on a very popular and widely used deep learning architecture (namely, U-Net), to which we apply Bayesian learning, leading to the so-called Bayesian U-Net. Building on the previous works on Bayesian deep learning [19–22], our goal here is to apply this to semantic segmentation. Bayesian deep learning relies on learning the distribution of the weights of the layers. The theoretical analysis is technically difficult and relies on several tricks (see Section 2.1.1 and [19–21]), but has been already implemented (e.g., Tensorflow probability [32]). In this work, we apply Monte Carlo Dropout (see Section 2.1.2), which is equivalent to Bayesian deep learning, but using traditional deep learning layers and optimization methods. We aim to show that a semantic segmentation can be obtained with pixel-wise confidence score. These confidence maps allow us to better understand why the network succeeds or fails in its prediction. It appears that this is likely to be related to inconsistencies in the database, which can be corrected thanks to the confidence maps. Finally, we showed that a Bayesian network learns the data distribution very accurately, and can achieve much better results than traditional networks when trained on an incomplete database, which can be useful for active learning.

The paper is structured as follows: in Section 2, the method of Bayesian learning is described (Section 2.1), the datasets are then introduced (Section 2.2), and, finally, the evaluation metrics are defined (Section 2.3). The results are presented in Section 3 and discussed in Section 4. Finally, the conclusion and perspectives are drawn in Section 5.

## 2. Material

In this section, we briefly present Bayesian learning, and explain why it is more interesting than traditional deep learning and how it is applied to neural networks. The architecture of our network, inspired by U-Net but involving Bayesian layers, is then introduced. The dataset, made of crops from a larger image, will be presented. Finally, we discuss the evaluation metrics used in our study, either standard ones in semantic segmentation or others tailored towards uncertainty estimation.

### 2.1. Method

Traditional Deep Learning methods are very efficient for semantic segmentation [7–11]. However, such methods are prone to produce over-confident decisions. On the one hand, it is easy to produce images (not recognizable to humans) that state-of-the-art Deep Neural Networks believe to be recognizable with high confidence [33]. On the other hand, a small change in the input image can lead to a very different prediction with high confidence [34].

Bayesian learning for CNN has been proposed recently [21] and is based on Bayes by Backprop [19,20]. It produces similar results to traditional deep learning methods. However, the weights of the network are no longer simple points but are sampled according to a distribution whose parameters are learned.

### 2.1.1. Bayesian Learning

Let us define function $f$ as $y = f(x)$ that, given inputs $X = \{x_1, \ldots, x_n\}$ and their corresponding outputs $Y = \{y_1, \ldots, y_n\}$, can produce predictive outputs. A prior distribution can be used over the space of functions $p(f)$, thanks to Bayesian inference. This distribution represents the prior belief of which functions are likely to have generated the data.

The likelihood, defined as $p(Y|f, X)$, describes the process by which, given a function, an observation is generated. The posterior distribution given the dataset, $p(f|X, Y)$, can be found thanks to the Bayes rule:

$$p(f|X, Y) = \frac{p(Y|f, X)p(f)}{\int p(Y|f, X)p(f)df} \tag{1}$$

A new output can be predicted for a new input data $x^*$ by integrating over all possible functions $f$:

$$p(y^*|x^*, X, Y) = \int p(y^*|f^*)p(f^*|x^*, X, Y)df^* \tag{2}$$

This equation is intractable due to the integration operation. We can approximate it by taking a finite set of random variables $\mathbf{w}$ (the weights of the layers) and condition the model on them.

Conducting a full Bayesian inference (in order to estimate the entire posterior distribution) would allow us to estimate the distribution of $f$. Bayesian inference adjusts the beliefs about a distribution in the light of data or evidence. The solution to Bayesian inference is to update the model through a process called variational inference. The idea behind the variational inference is to consider that the function $f$ is ruled by a finite number of parameters. In the case of deep learning, these parameters are the weights $\mathbf{w}$ of the model. The model that evaluates the function $f$ is ruled by a finite number of parameters $\mathbf{w}$. Variational inference allows us to evaluate the distribution $q(\mathbf{w})$ of each parameter $\mathbf{w}$.

In order to approximate the true posterior $q(\mathbf{w})$ of $\mathbf{w}$, a variational distribution $q(\mathbf{w}|\theta)$ of known functional form is used. In other words, the true posterior of $\mathbf{w}$ is approximated by a more simple distribution with parameters $\theta$ (e.g., a normal distribution $q(\mathbf{w}) \sim q(\mathbf{w}|\theta) = \mathcal{N}(\theta)$, with $\theta = (\mu, \sigma)$). The estimation of the parameters $\theta$ can be carried out by minimizing the Kullback–Leibler divergence between $q(\mathbf{w}|\theta)$ and the true posterior $p(\mathbf{w}|\{X, Y\})$ with respect to $\theta$. It can be shown [21] that the corresponding optimization objective or cost function can be written as:

$$\mathcal{F}(\{X, Y\}, \theta) = KL(q(\mathbf{w}|\theta)|p(\mathbf{w})) - \mathbb{E}_{q(\mathbf{w}|\theta)} \log p(\{X, Y\}|\mathbf{w}) \tag{3}$$

This is known as the variational free energy. The first term is the Kullback–Leibler divergence between the variational distribution $q(\mathbf{w}|\theta)$ and the prior $p(\mathbf{w})$ and is called the complexity cost. The second term is the expected value of the likelihood with respect to the variational distribution, and is called the likelihood cost. More details on Bayesian deep learning and variational inference can be found in [19–21].

### 2.1.2. Monte Carlo Dropout

A simple means of implementing Bayesian Deep Learning relies on Monte Carlo Dropout. It has been demonstrated by [22] that the Monte Carlo Dropout can be considered as a variational inference, with a variational posterior distribution defined for each weight matrix as:

$$\begin{aligned} z_i &\sim Bernoulli(p_i) \\ \mathbf{W}_i &= \mathbf{M}_i \cdot diag(z_i) \end{aligned} \tag{4}$$

with $z_i$ as the random activation or inactivation coefficients and $\mathbf{M}_i$ as the weight matrix before dropout is applied. $p_i$ is the activation probability for layer $i$ and can be learned or set manually. The idea is to consider dropout layers in the network and keep them active in both training and prediction.

### 2.1.3. Relation between Deep Learning, Bayesian learning and Monte Carlo Dropout

In traditional deep learning, models are conditioned on thousands (sometimes millions) of weights ($\mathbf{w}$) that are learned during training. Once learned, the weights are fixed for further inference.

In Bayesian deep learning, models are also conditioned on weights. However, we suppose that each weight follows an unknown distribution. This unknown distribution can be approached by a user-defined variational distribution (namely, $q(\mathbf{w}|\theta)$). Generally, this distribution $q$ is a normal distribution and $\theta$ denotes the two parameters of the normal distribution, i.e., the mean $\mu$ and the standard deviation $\sigma$. However, one can choose any variational distribution for the weights. Hence, unlike traditional networks, the weights of a Bayesian network are not fixed, but conditioned on the variational distribution, where parameters are fixed after the learning phase. Thus, the weights can have a wider range of values, allowing the model to more accurately learn the data distribution.

Monte Carlo Dropout is equivalent to Bayesian deep learning in terms of its advantages and drawbacks. Its main advantage is that it can be performed by using traditional deep learning optimisation methods (e.g., it is not necessary to add the Kullback–Leibler divergence in the cost function). The only condition is to have a learning layer (i.e., convolution layer or dense layer) followed by a dropout layer, active in both the training and prediction phases. The main drawback is the variational distribution; the user is unable to set the variational distribution. Thus, each weight can only have two values: 0 or a specific value learned during training. Although this seems limited, it is sufficient to more accurately learn the data distribution than a traditional network.

In order to obtain relevant results, several predictions need to be performed to explore a sufficient number of values for the weights.

### 2.1.4. Architecture

In this work, a fully convolutional neural network inspired by U-Net [5] is used. Let us note that we could have used a more recent framework as a baseline, such as the U-Net++ [35] or attention-based mechanisms, as proposed in [36], or even inception modules [37]. In fact, any deep learning model can be enhanced with Bayesian deep learning, the idea being to replace traditional layers (such as the convolution layer or dense layer) by a Bayesian layer (e.g., for convolution, a convolution layer with weights following a variational distribution that will be learned, or, for MCD, a traditional convolution layer followed by a dropout layer active in both the training and prediction phases). We decided to keep the simple (U-Net) approach in order to ensure that the obtained results are related to the Bayesian approach and not to the deep learning architectures that aim to improve the results. Since U-Net is a widely spread and simply replicated model, we expect the presented results to be reused more easily. Furthermore, we aimed to maintain a fully convolutional architecture, which is not compatible with attention-based mechanisms. In this paper, our goal is, therefore, not to provide a new semantic segmentation method, but to be able to provide confidence to the semantic segmentation.

The model is made of several convolution blocks (composed of convolution layers with ELU activation), followed by a pooling layer (when downsampling) or a deconvolution layer (when upsampling). After a pooling (resp. deconvolution), the number of filters of the convolution layers is multiplied (resp. divided) by two. Each upsampled output is concatenated with the output of the convolution block of the same size. A model, therefore, has three parameters: the block size (i.e., the number of convolutions in a block), the number of poolings, and the number of filters in the convolution layers of the first convolution block.

To produce uncertainty maps, the MC Dropout strategy is exploited. This is carried out by adding a dropout layer at the end of a convolution block. The dropout is active in both training and prediction.

The last convolution layer of the last convolution block has a number of filters corresponding to the number of classes and a softmax activation.

The network was trained using the Adam optimizer [38] with a batch size of 64 and an initial learning rate of 0.001. The learning rate is reduced on plateau (learning rate divided by 10 if no decay in the validation loss is observed in the 10 last epochs) and we also perform early stopping (stop the training if no decay in the validation loss is observed in the last 20 epochs). These are standard parameters, allowing the best results to be achieved while avoiding overfitting [39].

The architecture of the proposed model is presented in Figure 1.

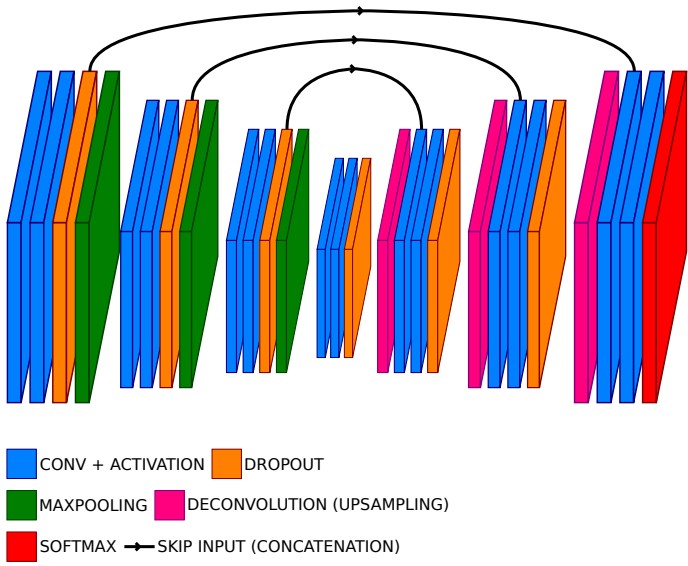

**Figure 1.** Architecture of our Bayesian U-Net (BU-Net) model with a block size of 2 and 3 poolings.

### 2.2. Dataset

To validate the proposed approach, we consider four different datasets representing various urban scenes. While the first three are public aerial datasets aiming to ease research reproducibility, the last one is a satellite dataset allowing us to demonstrate the behavior of our method on spaceborne imagery. We constructed our dataset by extracting small patches from original images. The training patches have an overlap of 25% (i.e., two separate patches may share 25% of common pixels). For each set (training and validation), we have two batches, with one corresponding to images of size $(N \times P \times P \times B)$ (where $N$ is the number of patches of size $P \times P$ extracted, $B$ is the number of bands in the input images). The batch corresponding to the ground truth is of size $(N \times P \times P \times C)$, where C is the number of classes.

The first dataset is derived from the Massachusetts Building dataset [40]. Small $64 \times 64$ patches are extracted from the images. The dataset is composed of images of size $1500 \times 1500$ pixels. Out of the 150 available images, 136 are used for training, 4 are used for validation and the 10 remaining images are for testing.

The second dataset is the Inria Aerial Image Labeling [41]. A total of 50 images were kept for training/validation (135 for training, 15 for validation), out of the 180 images of the dataset. Testing is performed on the 30 remaining images. The images were downsampled by a factor of 2; this allows for images with a spatial resolution of 0.6 m to be obtained (similar to the Toulouse and Massachussets dataset). Small patches of size $64 \times 64$ were extracted.

The third dataset is the ISPRS Vaihingen [42]: 80% of the images were used for training/validation, while the 20% remaining were kept for testing. Patches of size 128 × 128 were extracted. We decided to extract larger patches because of their higher spatial resolution. Having large patches also allows the model to more precisely learn the contextual information.

The last dataset is derived from a large Pleiades image of size 13,800 × 10,000, with four bands—red, green, blue, near-infrared 13,800—over the city of Toulouse (see Figure 2), France. The ground truth was derived from the database of the French Mapping Agency, and only 3 classes of interest (*Buildings*, *Vegetation* and *Water*) were considered, with the remaining classes being gathered in an *Others* class. A database of 64 × 64 patches from the image was created.

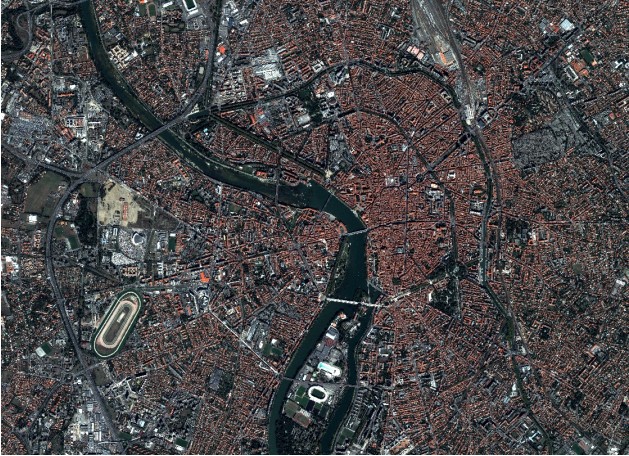

(**a**) Color view image.

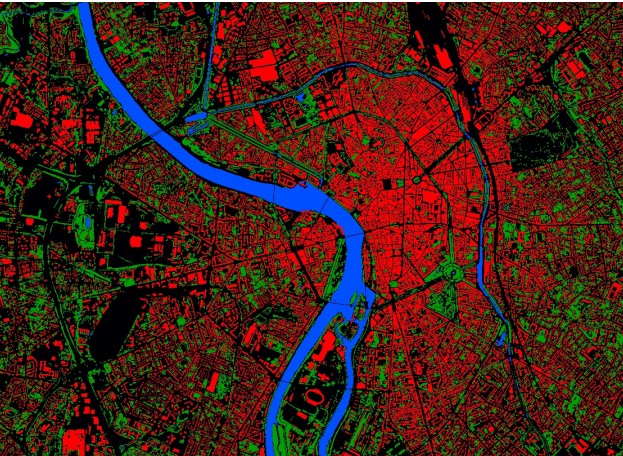

(**b**) Ground truth.

**Figure 2.** Toulouse dataset: a color view of the Pleiades image and its corresponding ground truth. Color code: 🔴 *buildings*, 🔵 *water*, 🟢 *tree*, ⚫ *background*.

### 2.3. Metrics

Once trained, a Bayesian model produces different predictions for the same input data, since its weights are sampled from a distribution. Therefore, several predictions need to be performed. At each iteration, the model will return a pixel-wise probability. The final semantic segmentation map is computed through a majority vote from all these predictions. One can then derive confusion matrices and the usual classification/segmentation quality metrics (precision, recall, accuracy, f-score, intersection over union (IoU) and kappa coefficient $\kappa$).

Although widely used, these metrics cannot assess the reliability of the model. Therefore, we also compute additional metrics in order to evaluate the level of uncertainty in the network.

First, measure the mean of the probability output for each class. This allows us to show how confident the network is in its prediction. To derive class uncertainty, the standard deviation of the probability outputs for each class has been computed.

Two other types of uncertainty measures are usually investigated. The Epistemic uncertainty, also known as model uncertainty, represents what the model does not know, due to insufficient training data. The Aleatoric uncertainty is due to noisy measurements in the data, and can be explained by increased sensor precision [21]. These two uncertainties, when combined, form the predictive uncertainty of the network. In this work, we derive two metrics, namely, the entropy of the predictive distribution (also known as predictive entropy) and mutual information between the predictive distribution and the posterior over network weights, as proposed in [23]. These metrics are very interesting, since mutual information captures epistemic (or model) uncertainty, whereas predictive entropy captures predictive uncertainty, which combines both epistemic and aleatoric uncertainties. The predictive entropy is computed as follows:

$$\hat{\mathbb{H}} = - \sum_c \left( \frac{1}{T} \sum_t p(y = c | x, \hat{w}_t) \right) \log \left( \frac{1}{T} \sum_t p(y = c | x, \hat{w}_t) \right) \tag{5}$$

where $c$ ranges over all the classes, $T$ is the number of Monte Carlo samples, $p(y = c | x, \hat{w}_t)$ is the softmax probability of input $x$ being in class $c$, and $\hat{w}_t$ are the model parameters on the $t^{\text{th}}$ Monte Carlo sample. The mutual information is computed as follows:

$$\hat{\mathbb{I}} = \hat{\mathbb{H}} + \frac{1}{T} \sum_{c,t} p(y = c | x, \hat{w}_t) \log(p(y = c | x, \hat{w}_t)) \tag{6}$$

To more precisely evaluate the impact of uncertainty metrics, we also compute some qualification maps. Our qualification map combines the validity of the majority vote (if the network predicted the right or the wrong label) and the uncertainty of the majority vote (how confident is the network). Two different color gradients describing the uncertainty of the prediction are used, for well- and wrongly classified pixels, respectively.

Furthermore, the uncertainty metrics can also be used to remove uncertain pixels. The refined segmentation map will contain some "unpredicted" pixels because these pixels are likely to have a wrong label if predicted. Conversely, the remaining predicted pixels are more likely to have the right label predicted.

### 2.4. Experiments

We conducted several experiments on the Massachusetts dataset to assess the robustness of MC Dropout models against different types of noise (see Figures 3 and 4). Three different scenarios have been explored, and for each, we compared our MC dropout model to a U-net baseline:

1.  Models are trained on the training samples of the dataset. The prediction is made on the test images on which a Gaussian noise of 5 dB (see Figure 3b) is added.
2.  Models are trained on the training samples of the dataset. The prediction is made on the test images on which a Salt and Pepper noise is added. In the images, 20% of the pixels are randomly selected and set to a value of 0 or 1 with an equal probability (see Figure 3c).
3.  Models are trained on a noisy database (a.k.a. label noise settings). In order to generate this noisy database, we randomly selected building polygons and set them to non-buildings. We iterate this process until at least 40% of building pixels have been swapped (see Figure 4).

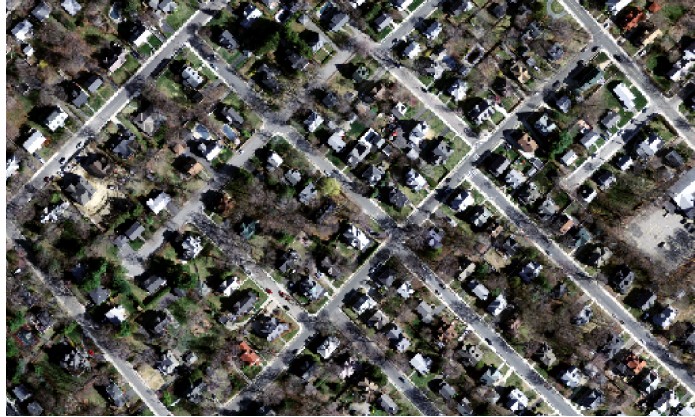

(**a**) RGB image without noise.

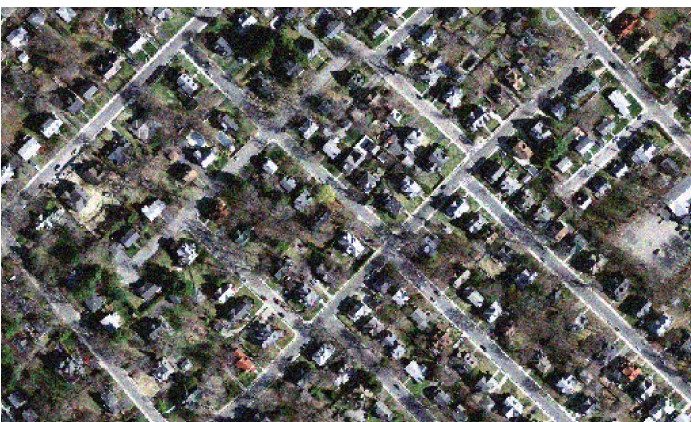

(**b**) Gaussian noise.

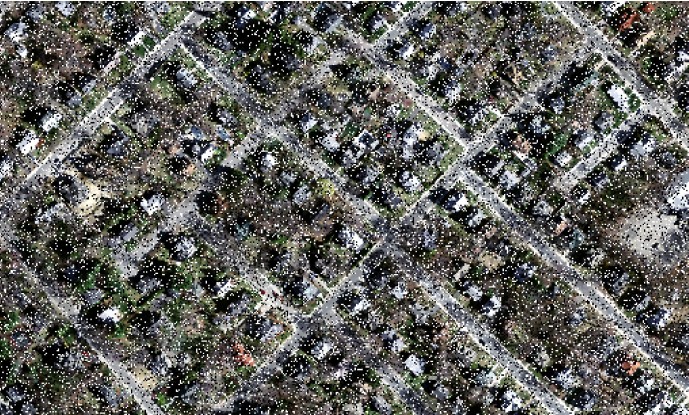

(**c**) Salt and Pepper noise.

**Figure 3.** Illustrations of noise added to the test images.

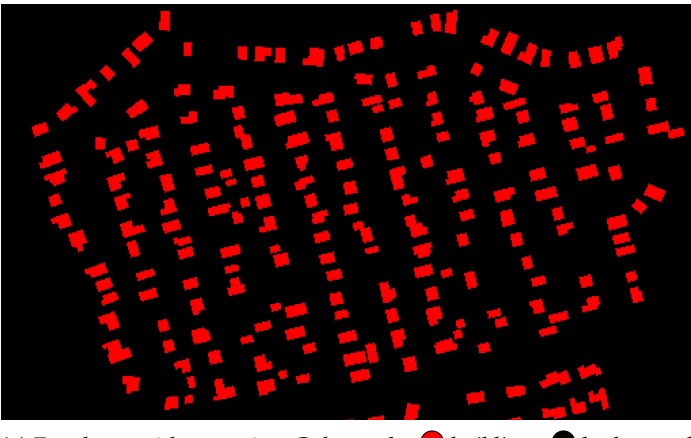

(**a**) Database without noise. Color code: ● *buildings,* ● *background*

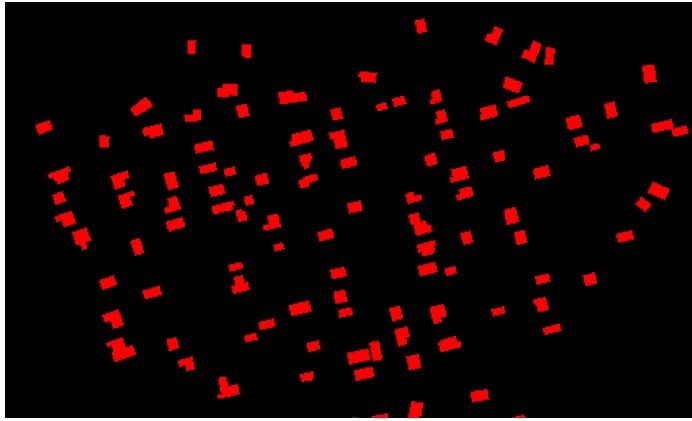

(**b**) Noisy database. Color code: ● *buildings,* ● *background*

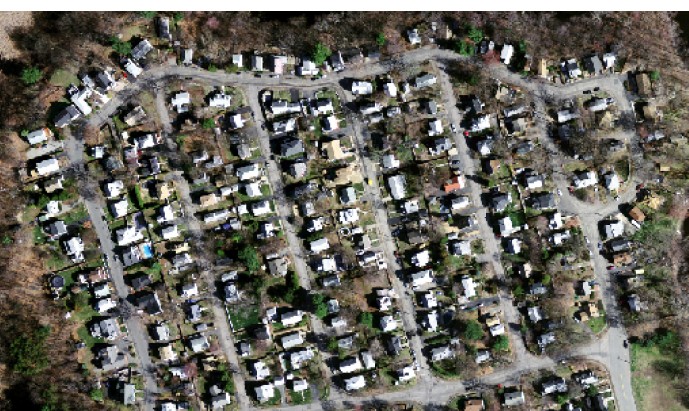

(**c**) RGB image corresponding to the noisy database

**Figure 4.** Illustration of label noise.

## 3. Results

### 3.1. Massachussets Dataset

The results for the Massachussets dataset are given in Figure 5 and Table 1. The mutual information was used for the computation on the qualification map. As one can expect from the state-of-the-art performances of deep networks, we achieve a high overall accuracy (86.26%) and F1-score (84.95%). For comparison in terms of F1-score, the method proposed by [43] achieveed 83.39% and the framework of [44] 84.72%. More interestingly, we also see the reliability of the uncertainty metrics on the test image (Figure 5d). When looking at the qualification map, one can see that some areas on the top right are in red, meaning

that the wrong label was predicted, but with high confidence. When taking a closer look at the input image and the ground truth, we can observe the error comes from the reference database (missing buildings in the ground truth). Thus, pixels with high confidence are likely to be correctly predicted. It is also interesting to point out that pixels from the class *background* are generally the most uncertain. This can be explained by the large variability of pixels in this class, leading to difficulties learning its proper distribution. In general, the higher the variability of the target class, the harder it is to learn the related distribution, leading to significant uncertainty.

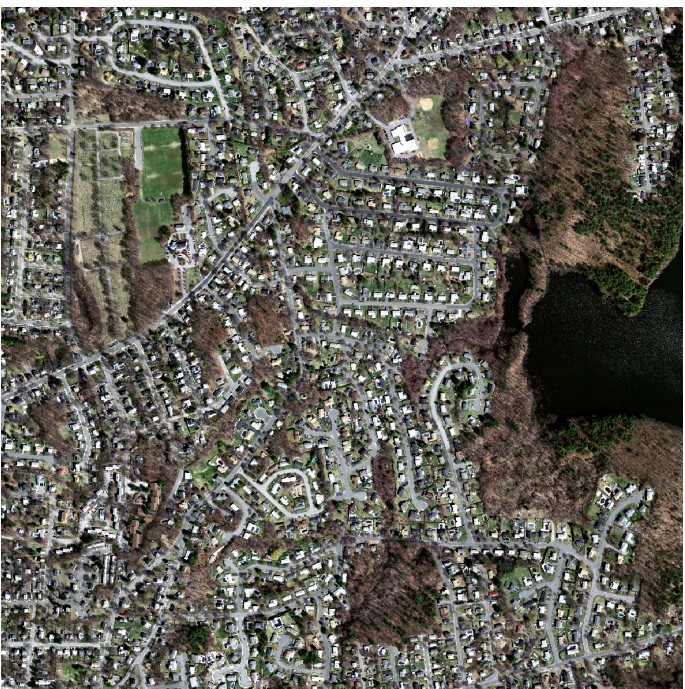

(**a**) RGB input image

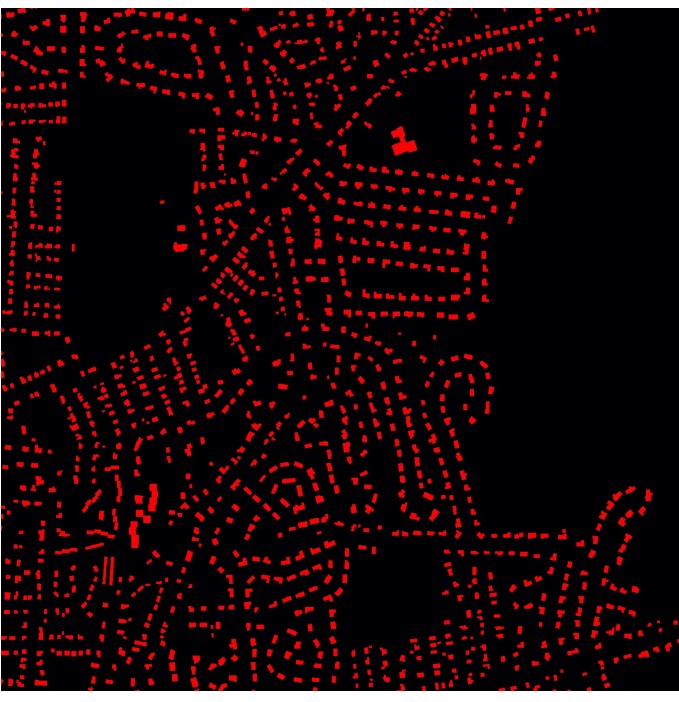

(**b**) Ground truth Color code: 🔴 *buildings*, ⚫ *background*

**Figure 5.** *Cont.*

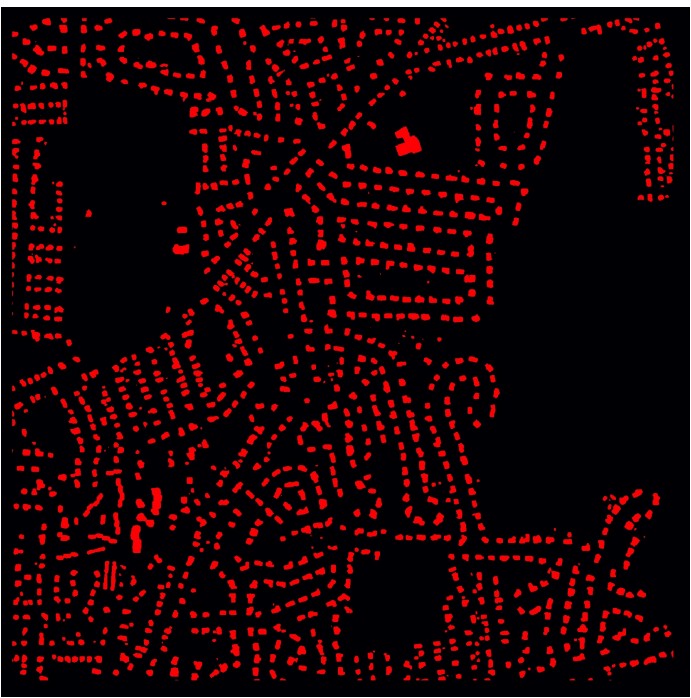

(**c**) Results of the semantic segmentation. Color code: ⬤ *buildings*,
⬤ *background*

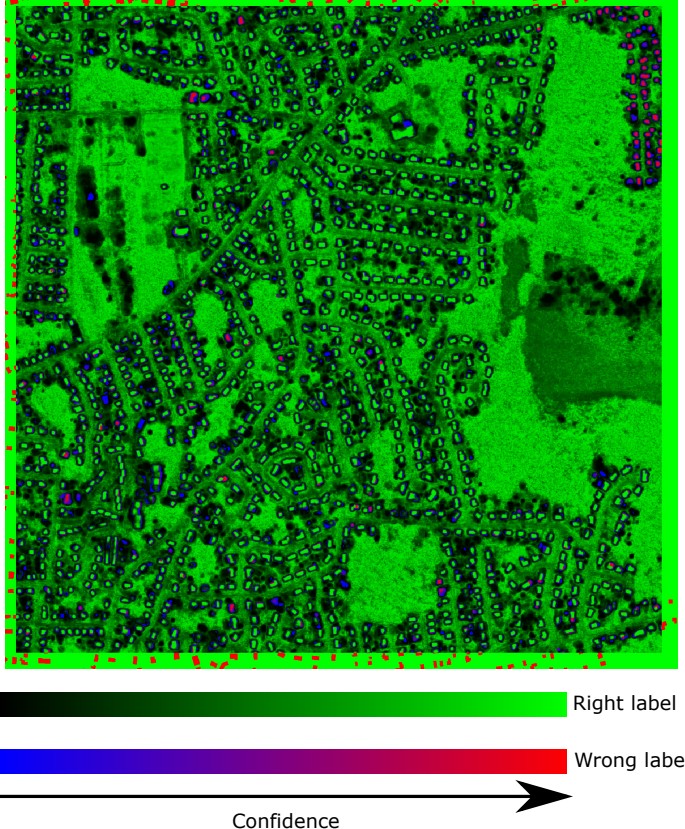

(**d**) Qualification map using the mutual information as an uncertainty metric

**Figure 5.** Visual results of our Bayesian U-Net on the Massachusetts dataset.

**Table 1.** Accuracy metrics of our Bayesian U-Net on the Massachussets dataset.

| Label | Background | Building | Overall |
|---|---|---|---|
| Precision | 59.34 | 95.69 | 77.52 |
| Recall | 82.85 | 87.04 | 84.95 |
| F-Score | 69.15 | 91.16 | 80.16 |
| IoU | 52.85 | 83.76 | 68.30 |
| Accuracy | 86.26 | | 86.26 |
| $\kappa$ | 0.61 | | 0.65 |

*3.2. INRIA Dataset*

The results of the INRIA dataset are given in Figure 6 and Table 2. Mutual information was used for the computation on the qualification map. Again, very high scores were achieved, with an overall accuracy of 93.63% and a F1-score of 88.94%). Similarly to the Massachussets dataset, we observe that wrong labels predicted with high confidence correspond to some missing buildings in the database. The class *background* reports a lot of pixels with the right label but low confidence.

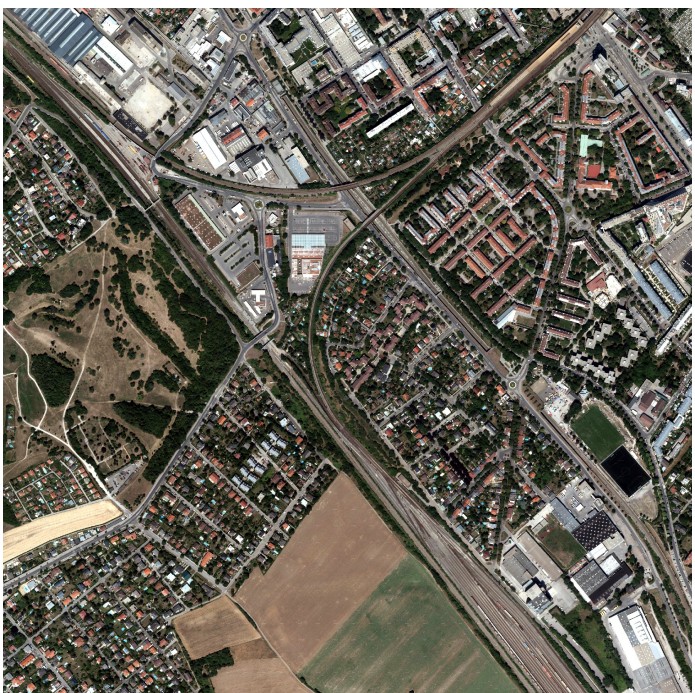

(**a**) RGB input image

**Figure 6.** *Cont.*

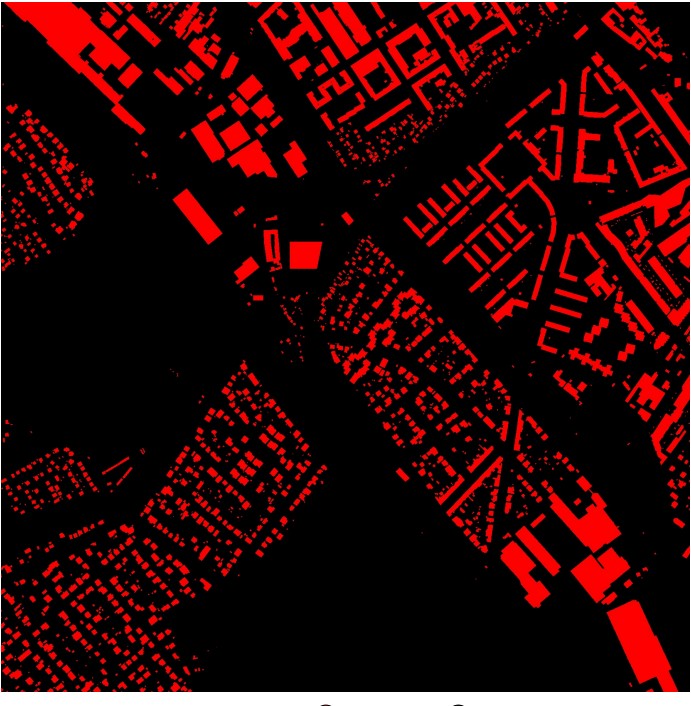

(**b**) Ground truth Color code: 🔴 *buildings*, ⚫ *background*

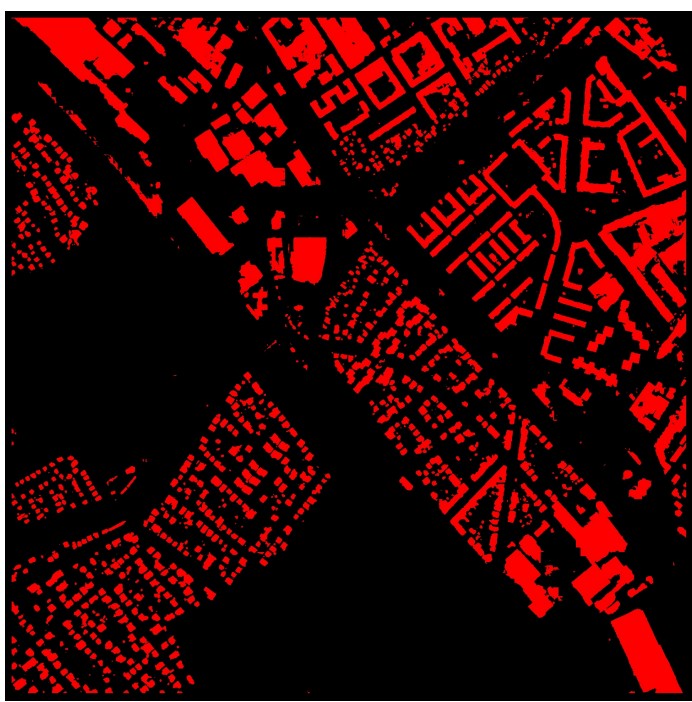

(**c**) Results of the semantic segmentation. Color code: 🔴 *buildings*, ⚫ *background*

**Figure 6.** *Cont.*

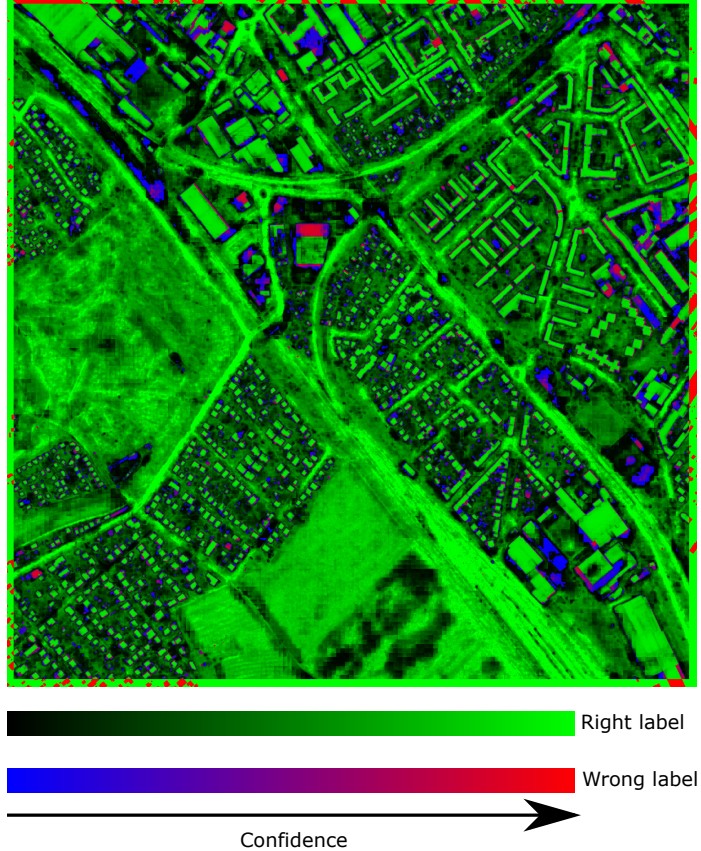

(**d**) Qualification map using the mutual information as an uncertainty metric

**Figure 6.** Visual results of our Bayesian U-Net on the INRIA dataset.

**Table 2.** Accuracy metrics of our Bayesian U-Net on the INRIA dataset.

| Label | Background | Building | Overall |
|---|---|---|---|
| Precision | 93.49 | 94.43 | 93.96 |
| Recall | 98.95 | 72.05 | 85.50 |
| F-Score | 96.14 | 81.74 | 88.94 |
| IoU | 92.57 | 69.11 | 80.84 |
| Accuracy | 93.63 | | 93.63 |
| $\kappa$ | 0.78 | | 0.93 |

### 3.3. ISPRS Vaihingen Dataset

The results obtained with the third public dataset, namely, ISPRS Vaihingen, are shown in Figure 7 and Table 3. The predictive entropy was used for computation on the qualification map. This dataset leads to the best results (overall accuracy of 93.22% and F1-score of 90.84%). The specificity of this dataset is the high similarity between the classes *low vegetation* and *tree*, especially when considering only the color information (let us recall that a digital surface model is also available, but not used here). Consequently, we observe some low confidence in these two classes (black pixels in Figure 7d). Furthermore, no pixels with the wrong label but high confidence were reported. This shows that our Bayesian U-Net is able to produce both a relevant semantic segmentation and uncertainty metrics. These metrics can be further used to derive a segmentation map with high confidence regarding the predicted pixels (see Section 4).

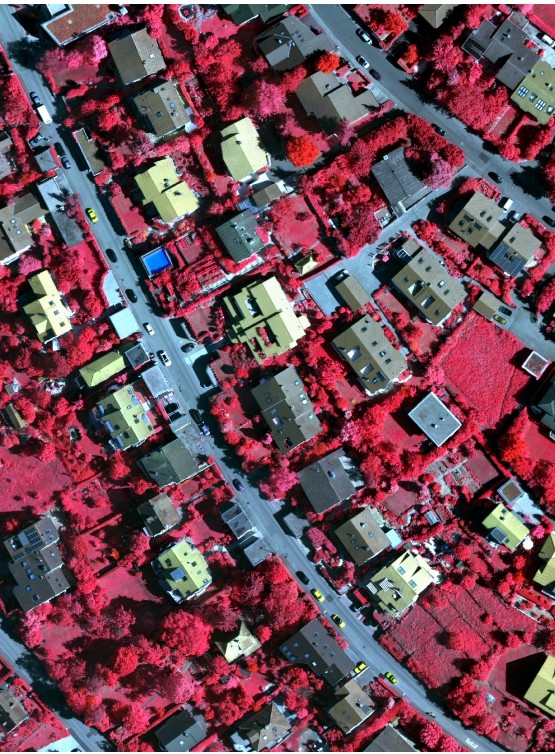

(**a**) False-color infrared image (NIR, red, green) input image

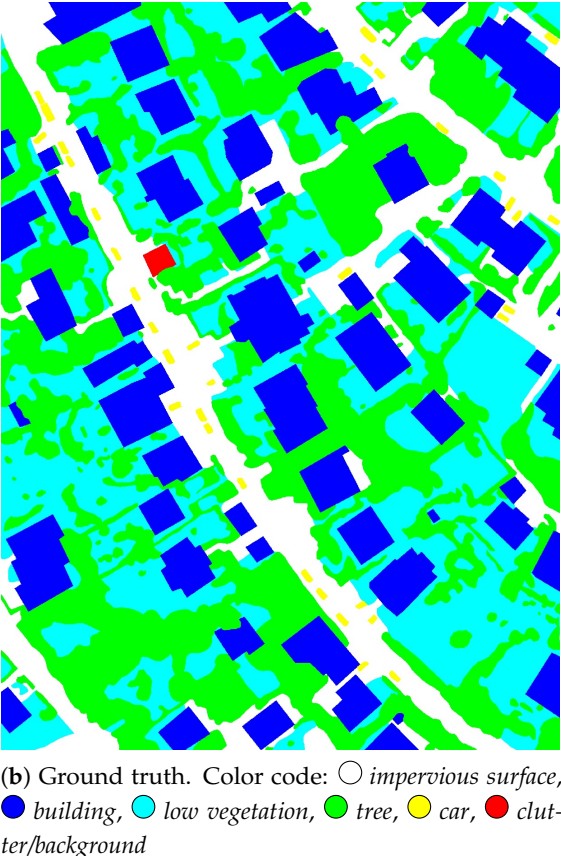

(**b**) Ground truth. Color code: ◯ *impervious surface,* 🔵 *building,* 🔵 *low vegetation,* 🟢 *tree,* 🟡 *car,* 🔴 *clutter/background*

**Figure 7.** *Cont.*

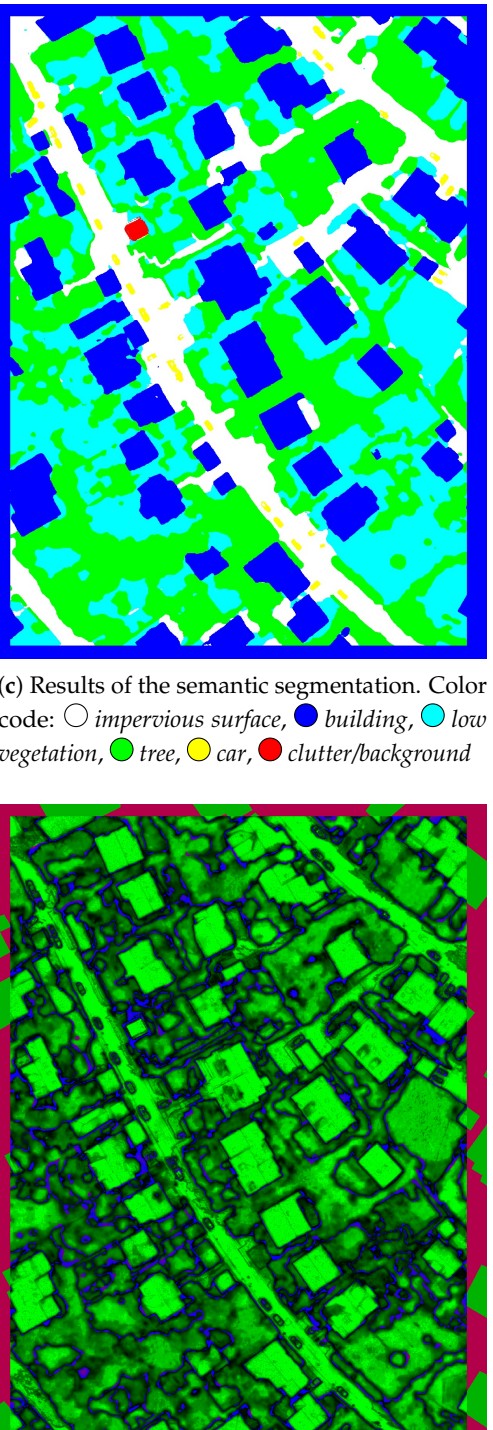

(**c**) Results of the semantic segmentation. Color code: ○ *impervious surface*, ● *building*, ○ *low vegetation*, ● *tree*, ○ *car*, ● *clutter/background*

(**d**) Qualification map using the predictive entropy as an uncertainty metric.

**Figure 7.** Visual results of our Bayesian U-Net on the ISPRS dataset.

**Table 3.** Accuracy metrics of our Bayesian U-Net on the ISPRS Vaihingen dataset.

| Label | Impervious Surface | Building | Low Vegetation | Tree | Car | Clutter/Background | Overall |
|---|---|---|---|---|---|---|---|
| Precision | 97.36 | 94.04 | 89.24 | 78.07 | 80.45 | 93.58 | 88.79 |
| Recall | 96.94 | 91.70 | 91.86 | 99.49 | 85.98 | 94.22 | 93.27 |
| F-Score | 97.15 | 92.85 | 90.53 | 87.49 | 83.12 | 93.90 | 90.84 |
| IoU | 94.46 | 86.66 | 82.70 | 77.76 | 71.11 | 88.50 | 83.53 |
| Accuracy | 98.78 | 95.00 | 94.84 | 99.96 | 99.76 | 98.10 | 93.22 |
| $\kappa$ | 0.96 | 0.89 | 0.87 | 0.87 | 0.83 | 0.93 | 0.93 |

### 3.4. Toulouse Dataset

Beyond the aerial imagery considered in the three previous datasets, we also assessed our Bayesian U-Net on Pleiades satellite imagery. More precisely, we tested our model on a 2048 × 2048 image of south-west Toulouse, France. This image was previously removed from the training set. Visual results are given in Figure 8, and quantitative measures are provided in Table 4. The mutual information was used for computation on the qualification map. The scores are somehow lower than with the other datasets, illustrating the challenges brought by satellite imagery. However, they could be considered satisfactory (overall accuracy of 80.77% and F1-score of 81.6%). More importantly, they contribute to the determination of the importance of a relevant training dataset. In some areas, classes may overlap; therefore, we decided to set such areas as a background. Indeed, when there are, for instance, roads under trees, the learned distribution is biased, leading to uncertainties in the tree class (see Figure 10 in Section 4).

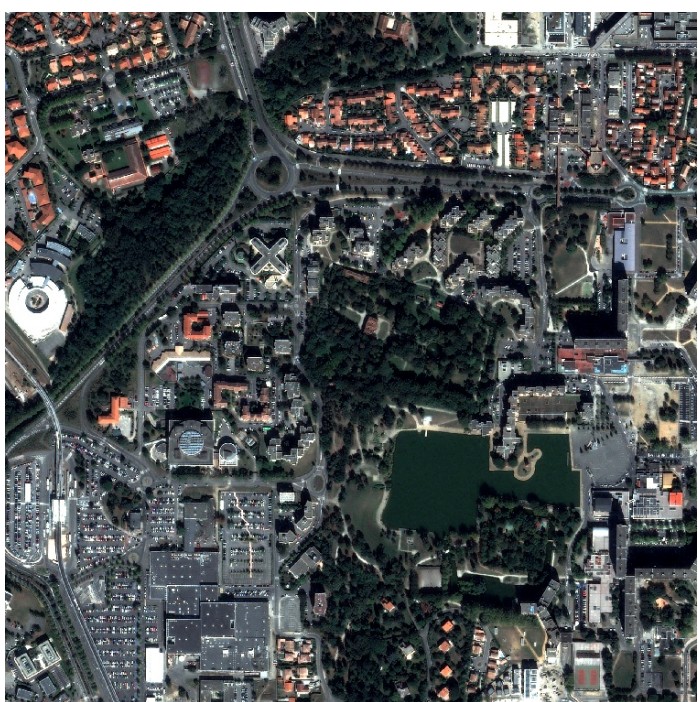

(**a**) RGB composition of the 4-bands optical image

**Figure 8.** *Cont.*

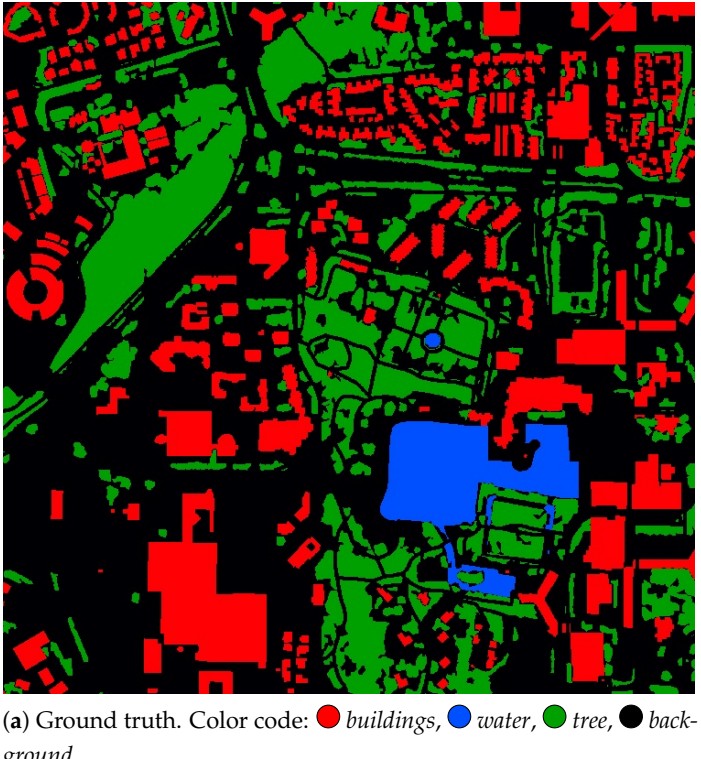

(**a**) Ground truth. Color code: 🔴 *buildings*, 🔵 *water*, 🟢 *tree*, ⚫ *background*

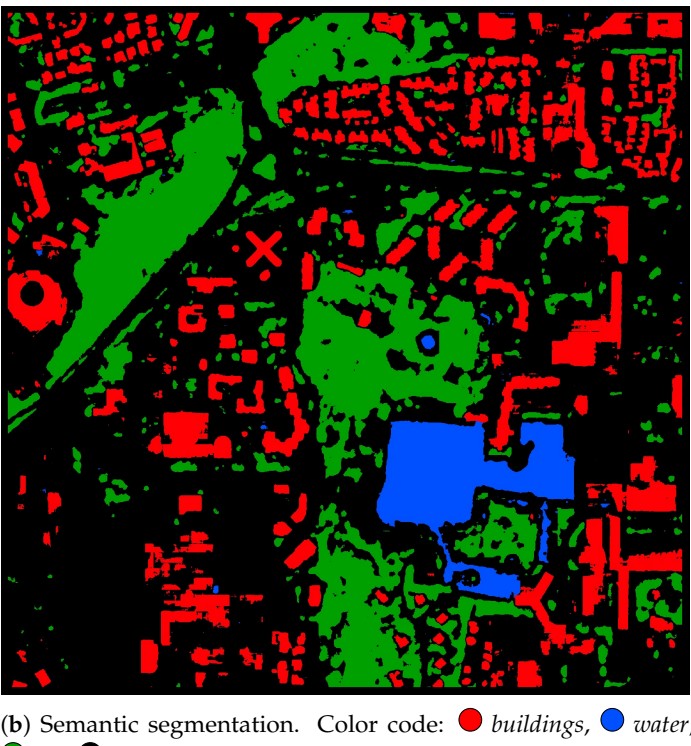

(**b**) Semantic segmentation. Color code: 🔴 *buildings*, 🔵 *water*, 🟢 *tree*, ⚫ *background*

**Figure 8.** *Cont.*

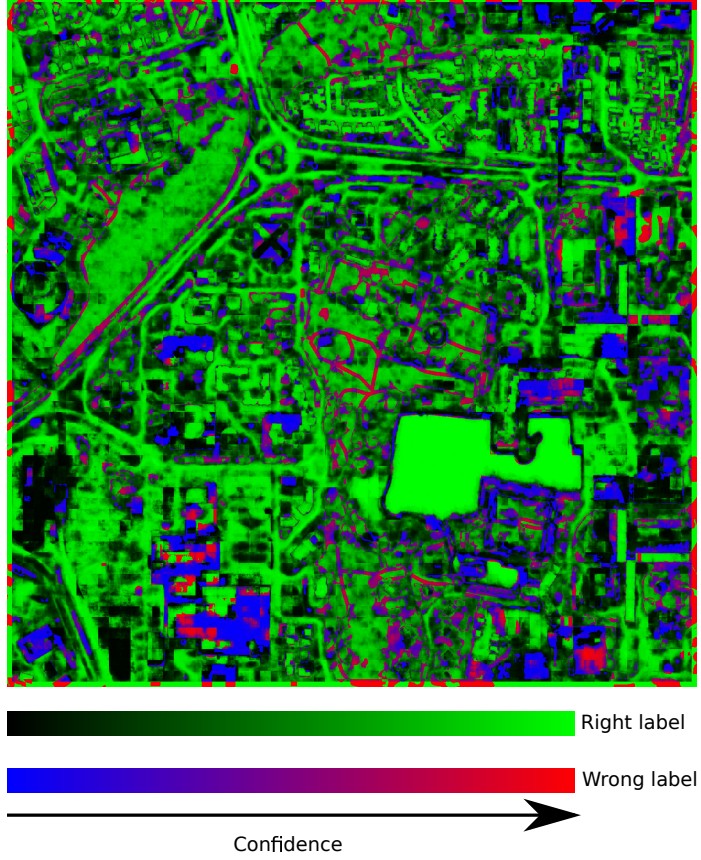

(**c**) Qualification map using the mutual information as an uncertainty metric

**Figure 8.** Visual results of our Bayesian U-Net on the Toulouse dataset.

**Table 4.** Accuracy metrics of our Bayesian U-Net on the Toulouse dataset.

| Label | Background | Buildings | Water | Vegetation | Overall |
|---|---|---|---|---|---|
| Precision | 83.67 | 74.01 | 95.45 | 75.05 | 82.80 |
| Recall | 84.66 | 85.01 | 89.76 | 66.61 | 81.51 |
| F-Score | 84.17 | 79.13 | 92.52 | 70.58 | 81.60 |
| IoU | 72.66 | 65.47 | 86.08 | 54.53 | 69.69 |
| Accuracy | 80.92 | 93.79 | 99.50 | 87.33 | 80.77 |
| $\kappa$ | 0.60 | 0.76 | 0.92 | 0.63 | 0.80 |

*3.5. Noise Robustness*

In our previous experiments, we assessed the performance of our Bayesian U-Net in perfect settings. The uncertainty estimation helped us to identify errors in the training set. Here, we conducted another round of experiments to study the behavior of our approach w.r.t. a baseline when facing different kinds of noise (as discussed in Section 2.4). Quantitative results are reported in Table 5. As expected, our Bayesian U-Net shows a higher noise robustness than a standard U-Net. When test images were corrupted with some Gaussian noise, both models gain precision, leading to a significant cost in terms of recall. It appears that both models are robust with respect to Gaussian noise, with accuracy metrics similar to standard training. Conversely, salt and pepper noise significantly lowers the quality scores, especially the precision. In this case, both models overpredict buildings, leading to a good recall at the cost of poor precision. The most significant difference concerns noisy labels. The traditional U-net struggles with very poor results, while our

Bayesian U-Net using MC dropout achieves results that are only slightly worse than those achieved the standard training set. This shows that our model not only provides uncertainty metrics but is also robust to noise. It can be trained on incomplete databases and still achieve good results.

**Table 5.** Comparison of the effect of different types of noise.

| Method | | Precision | Recall | F-Score | IoU | Accuracy | $\kappa$ |
|---|---|---|---|---|---|---|---|
| Standard Training | Baseline | 53.27 | 76.10 | 75.90 | 63.01 | 83.15 | 0.58 |
| | BU-Net | 59.34 | 82.85 | 80.16 | 68.30 | 86.26 | 0.65 |
| Gaussian noise (5dB) | Baseline | 54.13 | 71.54 | 75.54 | 62.58 | 83.06 | 0.57 |
| | BU-Net | 63.21 | 77.05 | 79.83 | 68.25 | 86.17 | 0.65 |
| Salt and Pepper noise | Baseline | 32.23 | 73.57 | 60.30 | 44.95 | 66.34 | 0.37 |
| | BU-Net | 35.98 | 86.55 | 64.03 | 48.49 | 68.87 | 0.43 |
| Label noise | Baseline | 33.61 | 42.39 | 60.43 | 47.27 | 73.72 | 0.30 |
| | BU-Net | 64.46 | 63.41 | 77.62 | 65.65 | 86.62 | 0.60 |

## 4. Discussion

The experimental results presented in the previous section show that our Bayesian U-Net, built using MC dropout, was able to achieve high-quality scores on several datasets. It also provided uncertainty metrics that are of high importance when assessing the behavior of the network. Indeed, from the qualification map, it can be observed that most of the wrongly predicted pixels are considered uncertain pixels (blue in the uncertainty maps). When looking more closely at the qualification maps, two main observations can be drawn:

- There are many areas that are well predicted but with low confidence (dark green and black areas). These areas correspond either to classes with a wide spectral variability such as the class *background* that gathers very different classes, e.g., vegetation, road, dirt, water (see Figures 5d and 6d) or the case when two classes are very close (e.g., *low vegetation* and *tree* in the ISPRS dataset, see Figure 7d).
- There are some areas where the predicted label is wrong, but the network is very confident in its prediction (red in the uncertainty maps). When taking a closer look at these areas, we can observe that, most of the time, they correspond to the label errors occurring in the reference database. In the Massachussets results, we can clearly see on the RGB image that some buildings are not referenced in the database, while other building footprints are incomplete (see Figure 9). Similar errors are reported in the Toulouse dataset. A main issue that arises when labeling airborne or spatial images is overlapping classes. In Figure 10, we observe, in the ground truth, that some walking paths are located within a small forest. One can imagine that there are, indeed, walking paths, but it is impossible to detect them in the image. Therefore, both results can be considered relevant: there are walking paths (labeled in the ground truth as *background*), but the image actually shows *trees*.

The second point is interesting because it allows us to correct a database by first inspecting the areas where the network predicted the wrong label with high confidence, before correcting labels for these areas using the input image. The qualification maps spot specific areas where a human user can modify the ground truth. These areas are likely to have the wrong label. This strategy probably leads to a more accurate ground truth without re-inspecting the entire image. Having such a refined ground truth would allow us to retrain the network, leading to a better description of the classes (less outliers) and also better accuracy (both in train, validation and test). Regarding the errors occurring in the original ground truth, one can assume that the evaluations of the results produced by the network are undervalued. Indeed, the model may have good prediction, but evaluation of the results on an incorrect database produces bad accuracy metrics.

To illustrate, a simple experiment was conducted on a small area of the Massachusetts test image: the test database was corrected by including *building* the pixels that were wrongly predicted but with a confidence greater than 50% (see Figure 11). A significant improvement was observed in terms of F1-score, meaning the detection (here focused on the *building* class) was more accurate. One can assume that if the ground truth was manually relabeled, the network will achieve better performances.

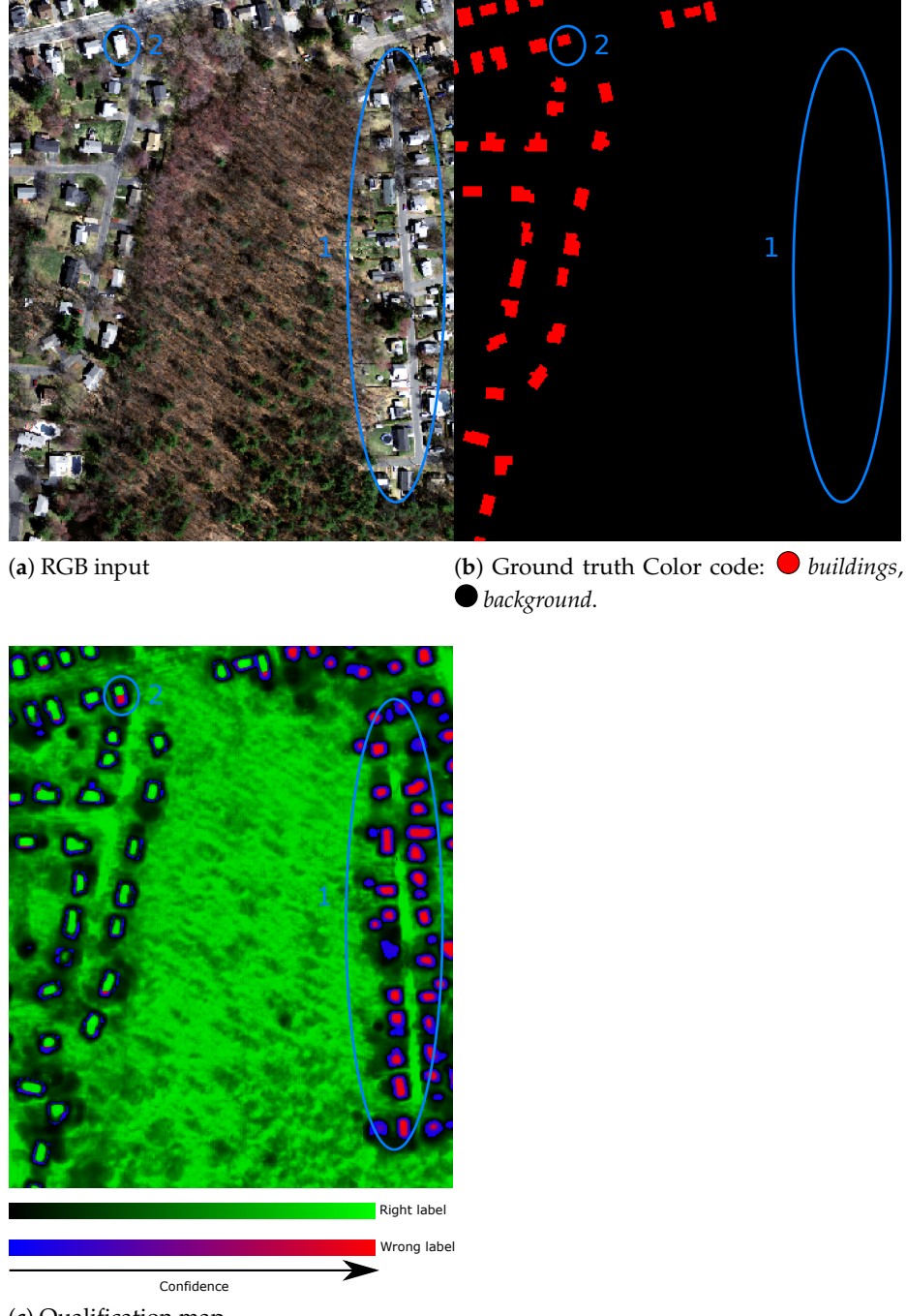

(**a**) RGB input

(**b**) Ground truth Color code: ● *buildings*, ● *background*.

(**c**) Qualification map

**Figure 9.** Visual results of our Bayesian U-Net on a small area of the Massachussets dataset. Blue circles correspond to: (1)-Building missing in the database, (2)-Incomplete building footprint.

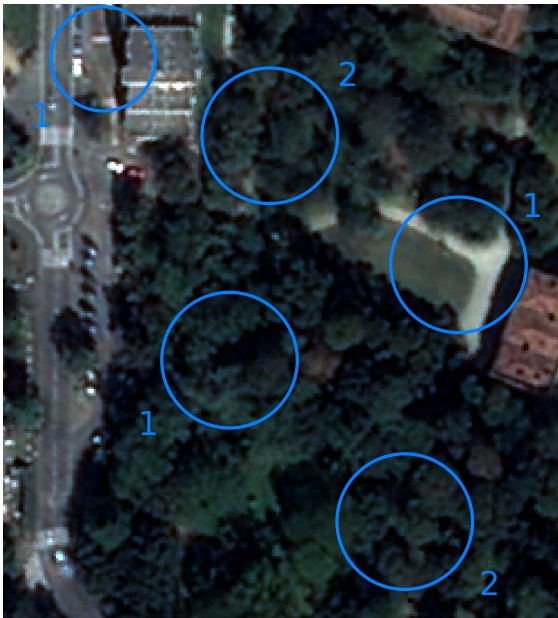

(**a**) RGB input

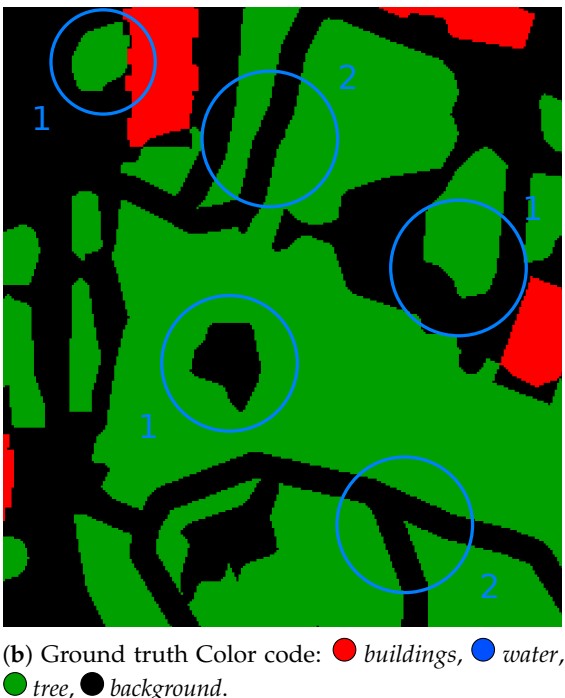

(**b**) Ground truth Color code: 🔴 *buildings,* 🔵 *water,*
🟢 *tree,* ⚫ *background.*

**Figure 10.** *Cont.*

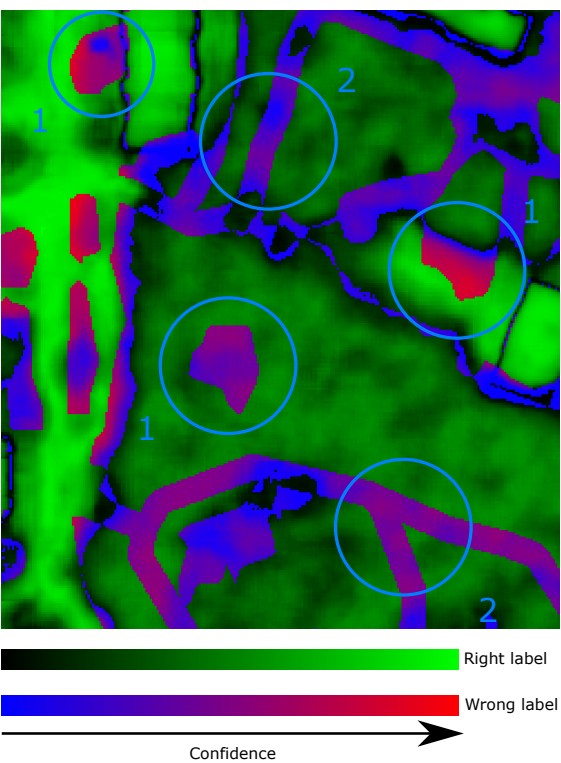

Right label

Wrong label

Confidence

(**c**) Qualification map

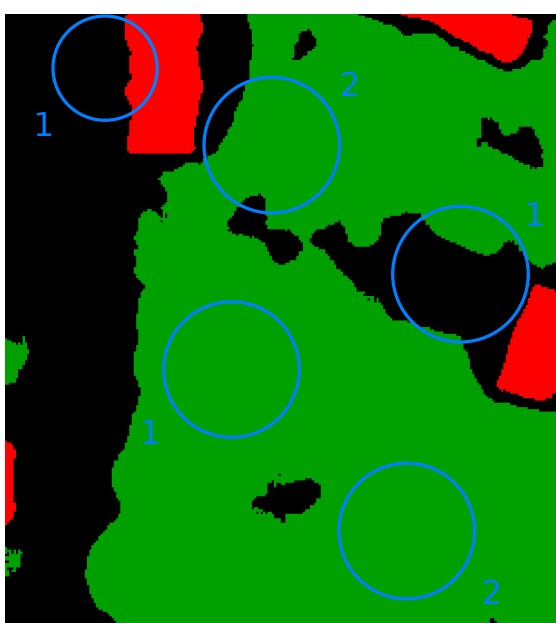

(**d**) Results of the semantic segmentation Color code: ● *buildings*, ● *background*

**Figure 10.** Visual results of our Bayesian U-Net on a small area of the Toulouse dataset. Blue circles correspond to: (1)-Wrong label in the database, (2)-Overlap between tree and walking path (labeled as *background* in the database).

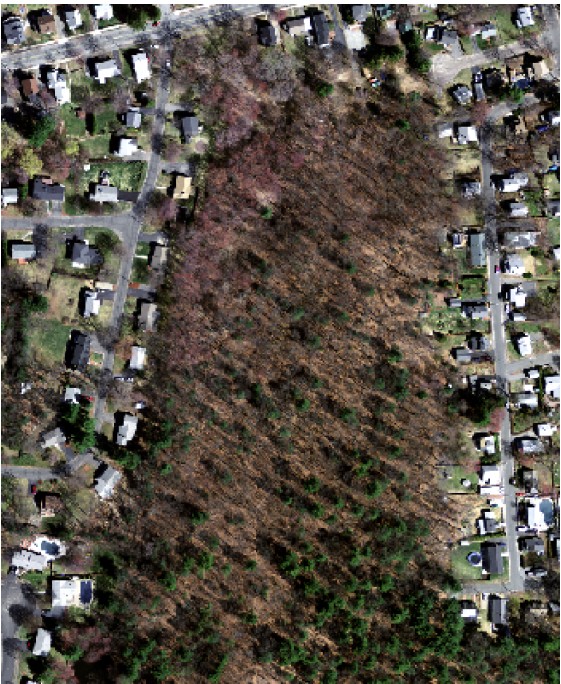

(**a**) RGB input

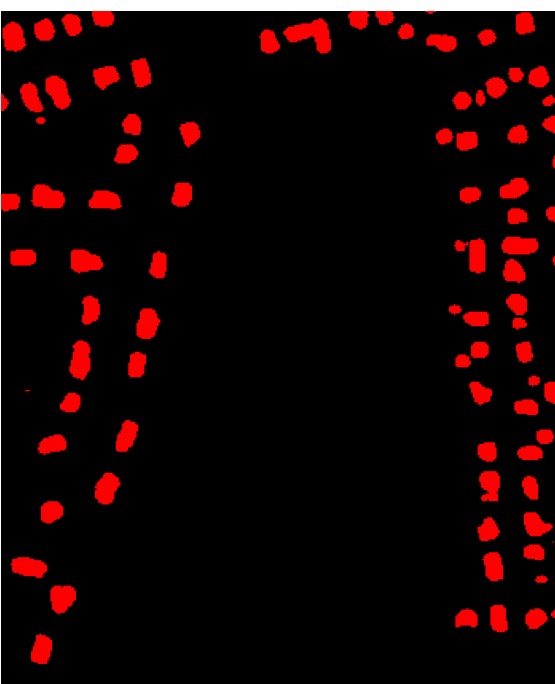

(**b**) Results of the semantic segmentation Color code:
🔴 *buildings,* ⚫ *background.*

**Figure 11.** *Cont.*

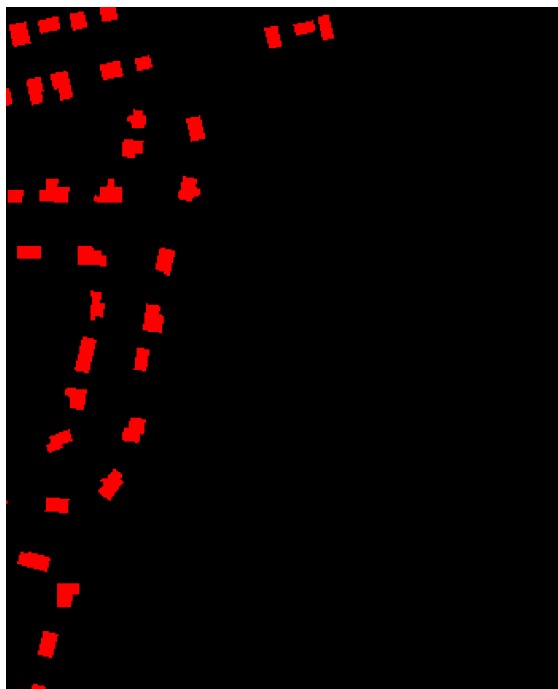

(**c**) Ground truth, reported accuracy: 95.02%, reported
F1-score: 75.45%

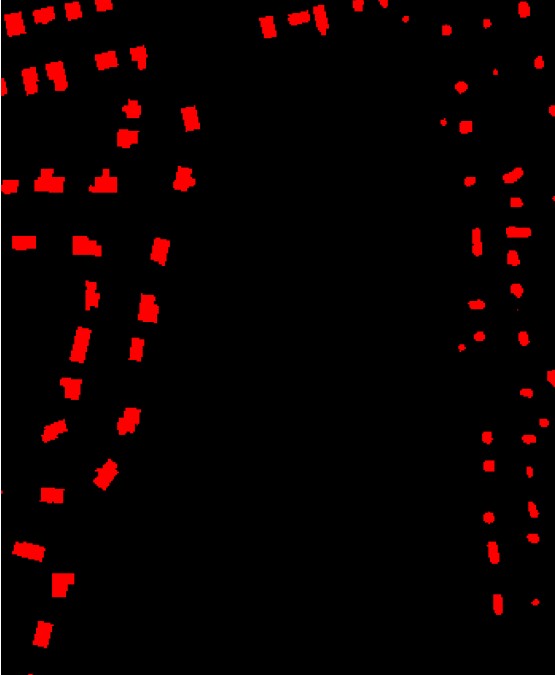

(**d**) Corrected ground truth, reported accuracy: 96.20%,
reported F1-score: 83.02%

**Figure 11.** Automatic correction of the Massachusetts database.

When no ground truth is available, it can be interesting to keep only pixels for which the prediction was performed with high confidence. Figure 12 shows segmentation results, considering that a percentage of the most uncertain pixels was removed. This can be used for various applications, e.g., target counting, since one can easily count and locate buildings in Figure 12d without having to know their precise footprints.

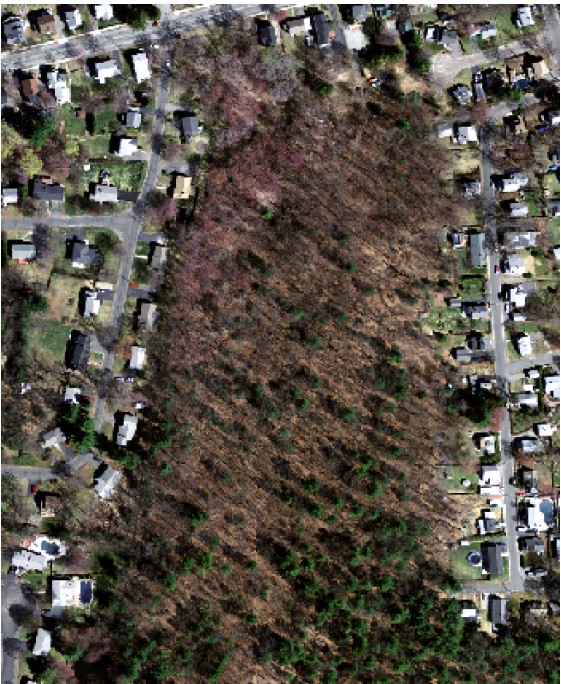

(**a**) RGB input

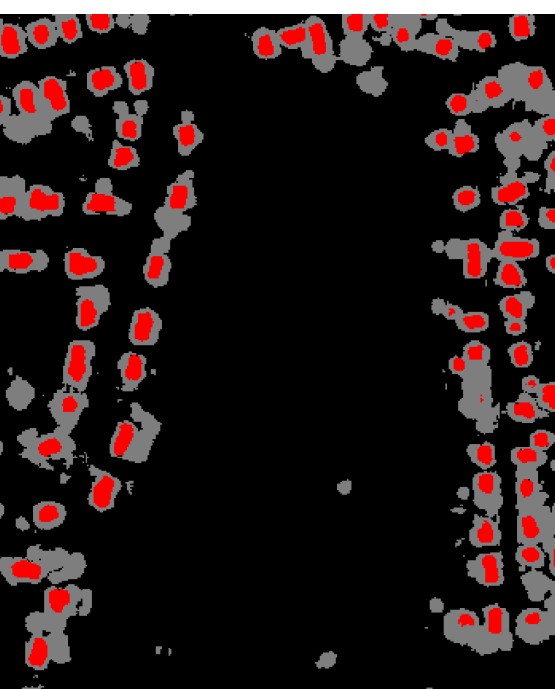

(**b**) Semantic segmentation with the 20% most uncertain pixels not predicted

**Figure 12.** *Cont.*

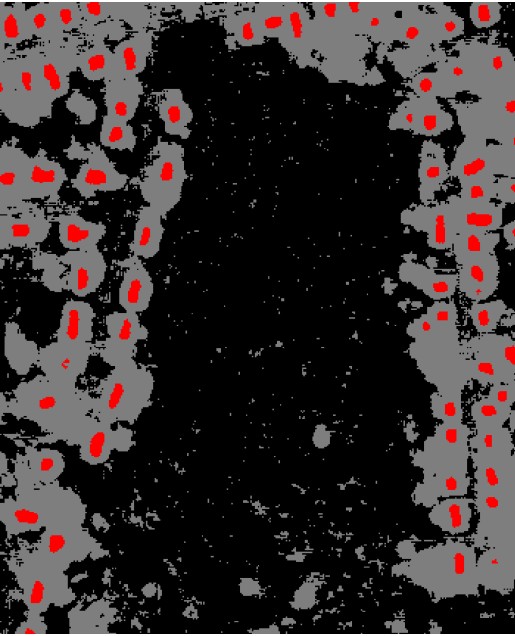

(**c**) Semantic segmentation with the 50% most uncertain pixels not predicted

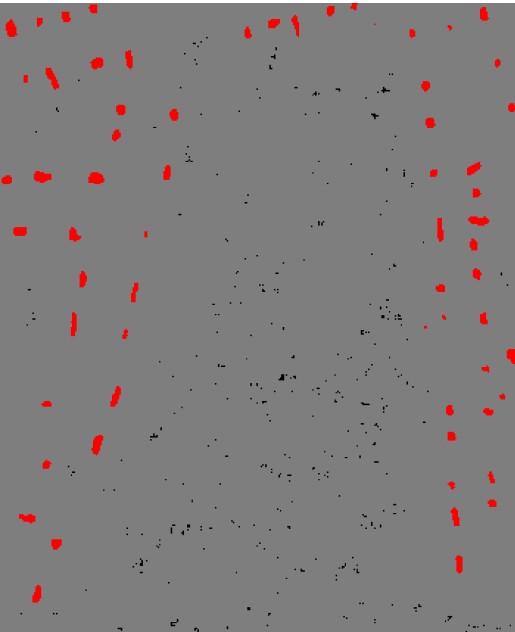

(**d**) Semantic segmentation with the 90% most uncertain pixels not predicted

**Figure 12.** Visual results of our Bayesian U-Net on a small area of the Massachusetts dataset. The pixels can be predicted with a desired level of confidence (here, greater than 20%, 50% and 90%). Color code: ● *buildings*, ● *background*, ● not predicted.

## 5. Conclusions and Perspectives

### 5.1. Conclusions

Deep learning methods have successfully been applied to the semantic segmentation of aerial and spatial optical images. Nevertheless, they are prone to overconfident results and still lack methods to estimate the uncertainty of the results. In this paper, we tackle this problem and propose a Bayesian deep learning model aiming to provide both a semantic segmentation and uncertainty maps. The model is based on the U-Net architecture that

can be tuned to a Bayesian model using Monte-Carlo Dropout. The proposed network performs well on several state-of-the-art datasets and can be applied to both aerial and spatial optical imagery. The obtained results are in line with state-of-the-art methods, with an F1-score ranging from 81.6% to 90.84% and overall accuracy ranging from 80.77% to 93.22%. Furthermore, Bayesian deep learning allows us to extract uncertainty maps that are very useful for qualification of the segmentation. Such maps, combined with the ground truth, may be useful to spot areas where the ground truth might be erroneous, leading to an automatic or a manual correction. In other words, the uncertainty maps could serve to produce a more accurate database. Training a deep network on such a refined database leads to improved performance. Uncertainty can also be used beyond a pure segmentation task, e.g., for object detection or counting. By removing uncertain pixels, it might be possible to detect the position of an object with high confidence. We illustrated this strategy with *buildings*, which are able to predict their location with high confidence, but not their precise shape.

### 5.2. Perspectives

Out future work would aim to evaluate how Bayesian deep learning can help to improve the quality of the ground truth. First, the areas where the network predicted the wrong label with high confidence need to be re-inspected and corrected if needed. Such re-labellisation would also allow us to assess how much we can rely on uncertainty metrics. The network would then be re-trained using the new ground truth. Finally, a comparison of the two networks, trained with their respective ground truths, would be made.

The second point is to investigate Bayesian neural networks with other variational inference methods. This would allow us to use a different distribution for the network weight (such as a normal distribution), since it tends to produce more significant uncertainty maps [30]. The main challenge here is the increase in number of parameters, leading to training complexity issues.

**Author Contributions:** Conceptualization, C.D., P.L. and S.L.; methodology, C.D.; investigation, C.D.; writing, review and editing, C.D., P.L. and S.L. All authors have read and agreed to the published version of the manuscript.

**Funding:** This research received no external funding.

**Institutional Review Board Statement:** Not applicable.

**Informed Consent Statement:** Not applicable.

**Conflicts of Interest:** The authors declare no conflict of interest.

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
