# Peer review of "Bayesian U-Net: Estimating Uncertainty in Semantic Segmentation of Earth Observation Images"

_remotesensing, doi:10.3390/rs13193836_

Round 1

Reviewer 1 Report

At the beginning the article sounds interesting however I noticed the following drawbacks:

  1. First of all, what is EO? The tile of the paper mentioned it but it never is explained.
  2. The word EO is only used 3 times along the paper. If the paper is related to this (non-defined) concept at least will be explained more.
  3. In the same context, what is CNN? and What is ISPRS?
  4. Different statements and paragraphs start with a reference, please check the writing style.
  5. What is the novelty of your paper?
  6. It is not clear who do you merge Bayesian Learning, Montecarlo and DL. Please provide a better explanation.
  7. The results need more discussion and information related to the DL procesess.

Author Response

At the beginning the article sounds interesting however I noticed the following drawbacks:

First of all, what is EO? The tile of the paper mentioned it but it never is explained.
EO stands for Earth Observation. This term is now explained in the revised manuscript: "Earth observation (EO) is the gathering of information about the physical, chemical, and biological systems of the planet Earth. Earth observation is used to monitor and assess the status of changes in natural and built environments.

The word EO is only used 3 times along the paper. If the paper is related to this (non-defined) concept at least will be explained more.
The term EO has been explained (see previous comment).

In the same context, what is CNN? and What is ISPRS?
CNN stands for Convolutional Neural Network. The ISPRS is the International Society for Photogrammetry and Remote Sensing. These two acronyms have been explained in the revised manuscript.

Different statements and paragraphs start with a reference, please check the writing style.
Following reviewer's suggestion, we have conducted a spell and style check of our manuscript (e.g. avoiding starting statements and paragraphs with a reference)

What is the novelty of your paper?
Our work proposes to apply Bayesian deep learning to semantic segmentation using the widespread U-Net architecture. We showed that Bayesian deep learning (using Monte Carlo Dropout) allows us to produce accurate semantic segmentation. In addition, we were able to derive pixel-wise confidence map that allows us to better understand why the network succeeds or fails on its prediction. It also appears that a Bayesian network (with MCD) learns more accurately the data distribution, even when the database is erroneous.

It is not clear who do you merge Bayesian Learning, MonteCarlo and DL. Please provide a better explanation.
Thank you for this suggestion. We added a section related to the link between these three concepts, namely Bayesian leaning, Monte Carlo and Deep Learning (see Section 2.1.3).

The results need more discussion and information related to the DL procesess. So far the results and discussion compose 9 pages out of the 22 pages for the whole manuscript. There are 8 Figures and 5 Tables that illustrate and describe our results. We assume it is enough to present our results. Nevertheless, we have added the IoU (Intersection over Union) metrics to our results. Shall the reviewer be more precise about the discussion / information that is lacking, we would be pleased to revise our manuscript accordingly.
Besides, we would like to emphasize that the DL processes are already detailled in Section 1.1.4; The network was trained using the Adam optimizer with a batch size of 64, an initial learning rate of 0.001. The learning rate is reduced on plateau (learning rate divided by 10 if no decay in the validation loss is observed in the 10 last epochs) and we also perform early stopping (stop the training if no decay in the validation loss is observed in the 20 last epochs). These are standard parameters, allowing to achieve the best results while avoiding over-fitting.

Reviewer 2 Report

The authors propose a  EO semantic segmentation works by coupling the most widespread architecture in the field (namely U-Net) with the Bayesian learning framework. However,  this paper is hard to understand due to the lack of theoretical analysis. In the reviewer's opinion, the current version of the paper is not acceptable for publishing due to poor presentation and needs further improvement.

1. The proposed model utilized the U-Net with the Bayesian learning framework, which are very common method. The authors should clarify whether this method is novel against the existing methods.
2. The theoretical analysis is too simple and lacks necessary explanation.
3. The presentation is poor including numerous typos and grammar errors throughout the manuscript, i.e. "unsing"->"using". The authors should improve the presentation to satisfy the publication standard.
4. The author should explain what EO is.

Author Response

The proposed model utilized the U-Net with the Bayesian learning framework, which are very common method. The authors should clarify whether this method is novel against the existing methods.
Our work proposes to apply Bayesian deep learning to semantic segmentation using the widespread U-Net architecture. We showed that Bayesian deep learning (using Monte Carlo Dropout) allows us to produce accurate semantic segmentation. In addition, we were able to derive pixel-wise confidence map that allows us to better understand why the network succeeds or fails on its prediction. It also appears that a Bayesian network (with MCD) learns more accurately the data distribution, even when the database is erroneous.

The theoretical analysis is too simple and lacks necessary explanation.
In this work, we apply Bayesian deep learning to semantic segmentation. Bayesian deep learning was introduced by others and we refer to their original papers for in-depth explanations (see references 19, 20, 21 and 22 of the revised version of our manuscript). As an introduction to Bayesian deep learning, we provide some material in Sections 2.1.1 and 2.1.2. We also added a Section (2.1.3) for a better understanding of this framework. Shall the reviewer identify what points were missing from the theoretical analysis, we would be pleased to revise our manuscript accordingly.

The presentation is poor including numerous typos and grammar errors throughout the manuscript, i.e. "unsing" $\rightarrow$ "using". The authors should improve the presentation to satisfy the publication standard.
Following reviewer's suggestion, we have conducted a full spell check of our manuscript.

The author should explain what EO is.
EO stands for Earth Observation. This term is now explained in the revised manuscript: "Earth observation (EO) is the gathering of information about the physical, chemical, and biological systems of the planet Earth. Earth observation is used to monitor and assess the status of changes in natural and built environments."

Reviewer 3 Report

This paper discusses interesting approach on EO Semantic Segmentation. Although has a partly novel contribution of uncertainty map dan Bayesian addition to a well known architecture UNet, this paper has several drawbacks: 1. The decision to choose the more dated U-Net architecture as the base of this research instead of the more recent U-Net++ [1] is a bit questionable. 2. Is there any strong reason not to use IoU or Jaccard Index as metric? Because F1 Score tend to give very high score for segmentation on a mediocre result. 3. The authors train and test their model on 3 separate datasets. These approaches tend to give overfit results. The authors should exclude a certain dataset and use it as test set only to prove that their model can generalize well. 4. More recent work uses attention for aerial images semantic segmentation [2]. I belief adding attention can improve the quality of this paper. 5. One small thing for future beginner reader, the authors should explain that EO is stands for Earth Observing, and how that term differ from aerial image, and satellite image. 1. Zhou, Zongwei, et al. "Unet++: A nested u-net architecture for medical image segmentation." Deep learning in medical image analysis and multimodal learning for clinical decision support. Springer, Cham, 2018. 3-11. 2. Niu, Ruigang, et al. "Hybrid multiple attention network for semantic segmentation in aerial images." IEEE Transactions on Geoscience and Remote Sensing (2021).

Author Response

The decision to choose the more dated U-Net architecture as the base of this research instead of the more recent U-Net++ [1] is a bit questionable.
We agree with the reviewer that a more recent U-Net++ could produce better results. But the goal of our study was rather  to show that a simpler model such as U-Net enhanced with Monte-Carlo Dropout would allow us to produce similar results to traditional U-Net with, in addition, some uncertainty metrics showing how confident is the network on its prediction. We also showed that such networks are very relevant when dealing with noisy labels without specific strategies. We have added a discussion about our choice of using the U-Net architecture in Section 2.1.4.

Is there any strong reason not to use IoU or Jaccard Index as metric? Because F1 Score tend to give very high score for segmentation on a mediocre result.
Thanks for the very useful comment. We thus added the IoU to our metrics.

The authors train and test their model on 3 separate datasets. These approaches tend to give overfit results. The authors should exclude a certain dataset and use it as test set only to prove that their model can generalize well.
In order to avoid overfitting, we follow a common procedure and perform early stopping. We agree with the reviewer that training on two datasets and evaluate the model on the third would show if the model generalize well. But such a strategy is more likely to be employed for domain-adaptation methods, which is not our purpose. Our strategy is to split a given dataset into train-validation-test sets that do not overlap. It is interesting to note that the INRIA dataset is already composed of images from different cities over the world (USA and Austria). Furthermore, the datasets used in this study are very different and as such, are not really compliant with the reviewer's suggestion. Indeed, the spectral bands of the images are different (e.g. the Massachusetts dataset has 3 bands, namely red, green and blue, while the ISPRS Vaihingen dataset also has 3 bands but with different wavelengths: near-infrared, red and green). To the best of our knowledge, considering running different experiments with different datasets (but not training on one, testing on the other) is the common practice for most of the semantic segmentation methods but the very few specifically addressing domain adaptation. 
Furthermore, our framework aims at predicting uncertainty: our objective is thus not to provide a novel semantic segmentation method that generalize well, but to provide a confidence score in addition to the semantic segmentation.

More recent work uses attention for aerial images semantic segmentation [2]. I belief adding attention can improve the quality of this paper.
As mentioned earlier, the purpose of our method is not to provide a new semantic segmentation method. We did not involve attention in our model since we aimed to keep the model fully convolutional. This allows us to predict large images even if the model has been trained on small patches. Following reviewer's comment, we now discuss in the revised manuscript the possibility to use attention in our model. Finally, let us emphasize that by sticking to the standard U-Net, we expect the paper to reach a larger audience and have a wider impact since it does not need to use a more recent (but less known) architecture.

One small thing for future beginner reader, the authors should explain that EO is stands for Earth Observing, and how that term differ from aerial image, and satellite image.
The term EO is now explained in the revised manuscript: "Earth observation (EO) is the gathering of information about the physical, chemical, and biological systems of the planet Earth. Earth observation is used to monitor and assess the status of changes in natural and built environments."

Zhou, Zongwei, et al. "Unet++: A nested u-net architecture for medical image segmentation." Deep learning in medical image analysis and multimodal learning for clinical decision support. Springer, Cham, 2018. 3-11. 2. Niu, Ruigang, et al. "Hybrid multiple attention network for semantic segmentation in aerial images." IEEE Transactions on Geoscience and Remote Sensing (2021).
We would like to thank the reviewer for the interesting references. They have been added and discussed in the revised manuscript.

Round 2

Reviewer 1 Report

Most of my comments have been addressed however the results need more discussion and information related to the DL processes. 

Author Response

We thank the reviewer for the provided feedback. As indicated in our previous response, we will be happy to provide more information on the DL process shall the reviewer’s request be more precise on the information missing in our paper.

Reviewer 2 Report

The authors present a  EO semantic segmentation works by  the Bayesian learning framework. The results are correctly presented and the conclusion summarizes the main outcomes of the research.

1. The introduction is quite interchangeable which would be fine if the authors had explained their own idea and method more clearly. Their claims are formulated in such a general way that it is hard to understand what they did exactly, for example how they apply Bayesian deep learning to semantic segmentation. Also, I think it would be good to mention that the Bayesian deep learning is a quite commonly used tool in  deep learning, what its usual problems are and how this was solved in this publication.

2. Although I find the basic idea of the publication reasonably novel, in total the paper is very hard to read and the improvements not convincing enough.

3. Since the implementation is done in coding the authors should provide all the software files needed to reproduce the results since that can contribute to other researchers in the same field to reproduce the experiments in an agile manner. This can be useful for future research initiatives on this topic.

Author Response

We thank the reviewer for the provided feedback. 

1. The introduction is quite interchangeable which would be fine if the authors had explained their own idea and method more clearly. Their claims are formulated in such a general way that it is hard to understand what they did exactly, for example how they apply Bayesian deep learning to semantic segmentation. Also, I think it would be good to mention that the Bayesian deep learning is a quite commonly used tool in deep learning, what its usual problems are and how this was solved in this publication.

R: In the previous revised version, we already stated that we are using  previous works on Bayesian deep learning. We added the following clarification on the revised manuscript:
"Bayesian deep learning relies on learning the distribution of the weights of the layers. The theoretical analysis is technically difficult and relies on several tricks (see Section 2.1.1 and [19–21]) but has been already implemented (e.g. Tensorflow probability [32]). In this work, we apply Monte Carlo Dropout (see Section 2.1.2), which is equivalent to Bayesian deep learning, but using traditional deep learning layers and optimization methods."
Since we perform Monte Carlo Dropout, the only difference with traditional deep learning is to keep the dropout layer active in both training and prediction, which is not difficult to perform.

2. Although I find the basic idea of the publication reasonably novel, in total the paper is very hard to read and the improvements not convincing enough.

R: In our study, we showed that a simple model such as U-Net enhanced with Monte-Carlo Dropout allows us to produce similar results to traditional U-Net with, in addition, some uncertainty metrics showing how confident is the network on its prediction. This could be used in the context of database correction or target counting (see Section 4.) We also showed that such networks are very relevant when dealing with noisy labels without specific strategies, which can be of great interest in order to produce relevant results on incomplete/incorrect databases. We hope such contributions will convince the reviewer about the value of our paper.

3. Since the implementation is done in coding the authors should provide all the software files needed to reproduce the results since that can contribute to other researchers in the same field to reproduce the experiments in an agile manner. This can be useful for future research initiatives on this topic.

R: We fully agree that providing the code to other researchers would be very valuable in order to reproduce the results. However, this issue is still under discussion at CNES (French Space Agency) and therefore cannot be done right now. Nevertheless, we hope that our code (or at least part of it) can be soon released to other researchers (but then since the code is yet not perfect for a user-friendly use, it would need some corrections and documentation).